# Toward High-Performance Electrochromic Conjugated Polymers: Influence of Local Chemical Environment and Side-Chain Engineering

**DOI:** 10.3390/molecules27238424

**Published:** 2022-12-01

**Authors:** Kaiwen Lin, Changjun Wu, Guangyao Zhang, Zhixin Wu, Shiting Tang, Yingxin Lin, Xinye Li, Yuying Jiang, Hengjia Lin, Yuehui Wang, Shouli Ming, Baoyang Lu

**Affiliations:** 1Department of Materials and Food, University of Electronic Science and Technology of China Zhongshan Institute, Zhongshan 528402, China; 2School of Materials and Energy, University of Electronic Science and Technology of China, Chengdu 610054, China; 3Jiangxi Key Laboratory of Flexible Electronics, Flexible Electronics Innovation Institute, Jiangxi Science & Technology Normal University, Nanchang 330013, China; 4School of Chemistry and Chemical Engineering, Liaocheng University, Liaocheng 252059, China

**Keywords:** electrochromism, flexible electrochromic device, asymmetric molecule, 5-fluorobenzo[c][1,2,5]thiadiazole

## Abstract

Three homologous electrochromic conjugated polymers, each containing an asymmetric building block but decorated with distinct alkyl chains, were designed and synthesized using electrochemical polymerization in this study. The corresponding monomers, namely T610FBTT810, DT6FBT, and DT48FBT, comprise the same backbone structure, i.e., an asymmetric 5-fluorobenzo[c][1,2,5]thiadiazole unit substituted by two thiophene terminals, but were decorated with different types of alkyl chain (hexyl, 2-butyloctyl, 2-hexyldecyl, or 2-octyldecyl). The effects of the side-chain structure and asymmetric repeating unit on the optical absorption, electrochemistry, morphology, and electrochromic properties were investigated comparatively. It was found that the electrochromism conjugated polymer, originating from DT6FBT with the shortest and linear alkyl chain, exhibits the best electrochromic performance with a 25% optical contrast ratio and a 0.3 s response time. The flexible electrochromic device of PDT6FBT achieved reversible colors of navy and cyan between the neutral and oxidized states, consistent with the non-device phenomenon. These results demonstrate that subtle modification of the side chain is able to change the electrochromic properties of conjugated polymers.

## 1. Introduction

Electrochromic conjugated polymers (ECPs) can easily display reversible optical absorption change accompanied with color change during the doping–dedoping process. For example, the ECPs can change colors by applying an electric field, accepting (reduction and dedoping process) or ejecting electrons (oxidation and doping process); thus, different absorption spectra switch between redox states [1,2,3]. ECPs have recently gained a significant amount of attention due to their good processability, outstanding mechanical flexibility, color adjustablity through structural modifications, fast response time, high optical contrast, etc. [4,5,6].

Among various existing ECPs, conjugated polymers containing donor–acceptor–donor (D-A-D) structures have achieved considerable success owing to their tunable bandgap and solid packing, rich color, and the possibility of obtaining high-performance electrochromic materials by deliberately changing their chemical structure [7,8,9,10]. The monomer with a D–A–D architecture is constructed with two main segments: a central electron-accepting core (A) and two electron-donating groups (D). Through changing D and A units, the D–A–D strategy not only improves the diversity of target molecules, but also avoids the poor solubility attributed to a strong aggregation of planar organic molecules [11,12,13,14,15]. For example, benzazole-EDOT bearing D–A–D-type ECPs can achieve a reversible color change between green and a highly transmissive state, accompanied with a 72% optical contrast during the redox process at a wavelength of 1500 nm [16]. The D–A–D-type ECP based on strong-electron-accepting-ability thiadiazolo [3,4-c]pyridine and 3,4-ethylenedioxythiophene (EDOT) exhibits a lower bandgap, higher coloration efficiency, and faster response time relative to its analogue based on benzo[c][1,2,5]thiadiazole [17]. EDOT-quinoxaline-EDOT electrochromic polymer possesses good electrochemical stability, with less than 8% charge loss after 5000 cycles [18]. D–A–D-type ECP containing isoindigo acceptor and EDOT donor exhibits a low bandgap and good electrochromic properties in the near-infrared region, including an optical contrast of 59%, response time of 0.5, and coloration efficiency of 362 cm^2^ C^−1^, which demonstrate that the D–A–D-type polymer would be a potential candidate as a near-infrared electrochromic material [19]. PolyCNDI, containing a naphthalene diimide acceptor, is a multielectrochromic polymer which possesses five colors under different redox states [20]. A 5,5′-Bibenzo[c][1,2,5]thiadiazole-based D–A–D-type ECP in the neutral state exhibited a red color and 40% optical contrast between the neutral state and oxidation state [21]. These results demonstrate that the simple synthesis route and good electrochromic properties make D–A–D-type ECPs the ideal choice for constructing electrochromic devices.

To further enhance the electrochromic properties of ECPs, an emerging type of ECPs with asymmetric D–A–D molecular topology has attracted significant attention of researchers [22,23]. Compared to their symmetric counterparts, asymmetric D–A–D exhibits a stronger intermolecular-binding interaction probably due to the increased dipole moment; hence, it induces the optical contrast and coloration efficiency enhancement of ECPs [24,25]. Asymmetric D–A–D-type ECPs consist of three categories: an asymmetric acceptor with symmetrical donors, an asymmetric acceptor with asymmetric donors, or a symmetrical acceptor with asymmetric donors [26,27,28]. Reynolds et al. reported a dual n-and p-type electrochromic device based on poly-(bisEDOT-PyrPyr-Hx2) that was extremely stable [29]. Zhang and colleagues synthesized two donor–acceptor–donor′ (D–A–D′) asymmetric ECPs, PSWE and PSWT, that outperformed symmetrical D–A–D PSWS in terms of optical contrast, response time, and coloration efficiency [30]. Toppare et al. presented asymmetric thiophene−benzothiadiazole−3, 4-ethylenedioxythiophene type ECP with a 1.18 eV bandgap, exhibiting p- and n-type doping superior to that with symmetrical analogues [31]. Therefore, the superior electrochromic properties of asymmetric D–A–D ECPs indicate great potential in electrochromic device applications.

In this contribution, we present three new asymmetric monomers, 7-(4-(2-hexyldodecyl)thiophen-2-yl)-5-fluoro-4-(4-(2-octyldodecyl)thiophen-2-yl)benzo[c][1,2,5]thiadiazole (T610FBTT810), 7-(4-hexylthiophen-2-yl)-5-fluoro-4-(4-hexylthiophen-2-yl)benzo[c][1,2,5]thiadiazole (DT6FBT), and 4-(4-(2-butyldecyl)thiophen-2-yl)-5-fluoro-7-(4-(2-butyldecyl)thiophen-2-yl)benzo[c][1,2,5]thiadiazole (DT48FBT) in Figure 1. All monomers employed an asymmetric unit of 5-fluoro-2,1,3-benzothiadiazole as the acceptor and thiophene derivatives bearing different alkyl chains as the donor, which provides an effective design strategy for asymmetric ECPs. Since fluorine possesses a strong electron-accepting ability, the introduction of a fluorine substituent has important influences on the properties of polymer film, such as intermolecular electrostatic interactions, polymeric film morphology, π–π stacking distance, charge transport ability, etc. [32,33]. Önal et al. reported 5-fluoro-2,1,3-benzothiadiazole-based ECPs [34,35,36], and analyzed the influence of a fluorine atom on the optical and electrical properties of a conjugated polymer. As we all know, alkyl side chains influence the redox behavior and electrochromic properties of the electrodeposited polymers [24,37]. It is essential to research the impact on electrochromic properties of polymers with subtle changes in the alkyl substituents of thiophene analogue donors. The designed asymmetric polymers, namely PT610FBTT810, PDT6FBT, and PDT48FBT, were prepared through the electropolymerization of asymmetric monomers (Figure 1). The effects of the side-chain structure and asymmetric repeating unit on the optical absorption, electrochemistry, morphology, and electrochromic properties were highlighted in detail.

## 2. Results and Discussion

### 2.1. Synthesis of T610FBTT810, DT6FBT, and DT48FBT

To comprehend the effect of an alkyl chain on D-A-D-type monomers, three D-A-D-type monomers were synthesized via a Stille cross-coupling reaction of 4,7-dibromo-5-fluorobenzo[c][1,2,5]thiadiazole and 2-(tributylstannyl)-4-alkylthiophenes. Importantly, T610FBTT810 showed different alkyl chains on the left and right thiophenes. All monomers were prepared at good yields of around 70% and displayed a bright orange color. The structure characterization of the monomers has been added in Appendix A.

### 2.2. Theoretical Calculations

The ground-state-optimized molecular geometries and frontier molecular orbital distributions of T610FBTT810, DT6FBT, and DT48FBT were carried out using density functional theory (DFT) using Gaussian 09 at the B3LYP/6-31G(d) level, as shown in Figure 1. T610FBTT810, DT6FBT, and DT48FBT had slight twisted structures with dihedral angles of 10°, 0° and 8° when the sulfur atoms of the benzothiadiazole unit and thiophene unit are at the *cis* position. Meanwhile, all monomers were found to be planar at the *trans-cis* position. In this respect, although alkyl chains showed steric hindrance, N···S and F···S noncovalent interactions, namely, “conformational locks”, can fabricate a planar backbone [38]. Therefore, all monomers would be high-mobility semiconductors, promoting crystallization and facilitating charge transfer [39]. For all monomers, the electron density distribution of the lowest unoccupied molecular orbitals (LUMOs) and the highest occupied molecular orbitals (HOMOs) localized in the entire conjugated skeleton, which is favorable to the polymerization of D-A-D-type monomers at a low applied potential. The values of the dipole moment and HOMO and LUMO gaps exhibited no significant changes with different alkyl chains, which are shown in Figure 1.

### 2.3. Electrochemistry

The anodic scan for T610FBTT810, DT6FBT, and DT48FBT was carried out in CH_2_Cl_2_-Bu_4_NPF_6_ (0.1 mol L^−1^) containing 10 mmol L^−1^ monomer to explore the effect of molecular structure on oxidation behavior (Figure 2). The onset oxidation potentials (*E*_onset_) were decreased along with the increase in the alkyl chain. The *E*_onset_ is summarized as: 1.28 V for DT6FBT, 1.19 V for DT48FBT, and 1.14 V for T610FBTT810 vs. Ag/AgCl. The higher oxidation potential of DT6FBT relative to DT48FBT and T610FBTT810 is attributed to the weaker electron-donating ability of the hexyl chain than the 2-hexyldecyl and 2-octyldecyl chains.

Firstly, the polymerization of T610FBTT810, DT6FBT, and DT48FBT was carried out using cyclic voltammetry (CV) that is a popular method to investigate the redox property of as-prepared polymer films. As shown in Figure 3, the current density increased along with the growth of the cycle number corresponding to cyclic voltammograms (CVs), indicating polymers with good electrochemical activity were deposited on the working electrode [40]. Meanwhile, the polymer film coating on the working electrode could be observed. Additionally, the three polymers exhibited obvious redox couples: the reduction potential shifted to a higher potential along with the increase in alkyl chain lengths (0.8 V for DT6FBT, 1.0 V for DT48FBT, and 1.1 V for T610FBTT810 vs. Ag/AgCl). Additionally, the obvious redox potential shift of the polymer could be observed during the polymerization process of the monomer. The phenomenon results from the fact that the polymers possess increasing electrical resistance along with the growth rate of polymers, which has to be balanced by the additional applied potential [41,42].

Additionally, the electrochemical behaviors of PT610FBTT810, PDT6FBT, and PDT48FBT were investigated using CV in monomer-free CH_2_Cl_2_-Bu_4_NPF_6_ (0.1 mol L^−1^). The polymers were prepared using the potentiostatic method at a constant potential of 1.50 V for DT6FBT, 40 V for DT48FBT, and 1.35 V for T610FBTT810 with a charge of about 5 mC. As shown in Figure 4, the CVs of polymers at different scan rates were recorded in the potential windows of 0.4~1.4 V (PDT48FBT and PDT6FBT) and 0.7~1.4 V (PT610FBTT810). The three polymers showed a distinct potential drift, with the corresponding reduction peaks shifting in a negative direction as the scan rate decreased, demonstrating that this was a diffusion-controlled process, which has been observed in previously reported ECPs and can be explained by the following reasons: the slow transformation of conjugated blocks between the aromatic structure and the quinoid structure, the local rearrangement of polymer chains, slow heterogeneous charge transport, etc.

### 2.4. Optical Property

UV-vis spectra of monomers (dissolved in 0.1 M CHCl_3_) are illustrated in Figure 5A. All monomers exhibited comparable characteristic absorbance peaks of 252 nm, 308 nm, and 450 nm, which were attributed to the thiophene derivative donor unit, 5-fluoro-2,1,3-benzothiadiazole acceptor unit, and intramolecular charge transfer (ICT) between D and A units. The almost overlapping absorption peaks indicated that the solubility of the monomers was desirable with no aggregation. Additionally, the fluorescence emission spectra of T610FBTT810, DT6FBT, and DT48FBT in 0.1 M CHCl_3_ with the same excitation wavelength of 450 nm are shown in Figure 5B. In accordance with the UV-vis spectra, T610FBTT810, DT6FBT, and DT48FBT displayed comparable emission peaks at 555 nm. These results demonstrate that the slight length variation of alkyl chain structures have little influence on the optical properties of a monomer.

### 2.5. Spectroelectrochemistry

As the potential increased, the conjugated block of electrochromic polymer turned from an aromatic structure into a quinoid structure, accompanied by the optical absorption change [1,2]. Therefore, the spectroelectrochemical performance of the three polymers was measured by recording its optical absorption change under different applied potentials. The spectroelectrochemistry of PDT6FBT is given as an illustrative example in Figure 6. The neutral PDT6FBT possesses two distinct absorbance peaks located at 330 and 558 nm. The optical bandgap (*E*_g_^opt^) of PDT6FBT was calculated to be 1.63 eV according to the formula: *E*_g_^opt^ = 1240/*λ,* where *λ* is the edge absorption spectra of PDT6FBT (762 nm). As the applied potential increased, both absorption bands of PDT6FBT underwent blue shift along with the increase of 330 nm in the absorbance peak and a decrease of 558 nm in the absorbance peak. Meanwhile, a new absorption band located at 760 nm began to increase gradually, which resulted from the formation of the polaron and bipolaron [5,6]. During the oxidation process, PDT6FBT displayed a distinct color change from navy (neutral state; L: 34.64, a: 4.35, b: −10.49) to cyan (oxidation state; L: 7.45, a: 0.42, b: −0.60). The recorded absorption spectra of PDT6FBT passed through an isosbestic point at 640 nm, demonstrating easy mutual transformation between its neutral and oxidized states [1,2]. PT610FBTT810 and PDT48FBT in this system, because of the long alkyl chain, were soluble as soon as they were polymerized on the ITO, which made it difficult to obtain the suitable films. Therefore, the absorption spectra variations were unobserved under different potentials (Appendix A).

### 2.6. Electrochromic Performance

The kinetic study of PDT6FBT was characterized by a time–transmittance curve in monomer-free CH_3_CN-Bu_4_NPF_6_ (0.10 mol L^−1^) solution, as shown in Figure 7. According to the results, the electrochromic parameters, including optical contrast ratio (Δ*T*), response time, and coloration efficiency (*CE*), were obtained and are summarized in Table 1.

The time–transmittance curves and electrochromic parameters of PDT6FBT are given as an illustrative example in Figure 7 and Table 1. PDT6FBT displayed a moderate optical contrast ratio at different wavelengths, fast response time of oxidation and reduction processes, and decent *CE* values. The Δ*T* hardly changed at 380 nm, 550 nm, and 750 nm, ranging from 23% to 25% (10 s intervals for 200 switches). In addition, in order to characterize the effect of different switching time on its electrochromic performance, we also studied the time–transmittance curves of PDT6FBT with switching times of 2 s, 5 s, and 10 s, respectively (Appendix A ). With the decrease in the switching time, the optical contrast of the PDT6FBT also showed a different tendency to decrease. The short wavelength of PDT6FBT (380 nm) exhibited an obvious reduction from 23% to 15% with switching time from 10 s to 2 s. 550 nm of PDT6FBT exhibited a slight reduction from 25% to 21%. Long wavelength maintained the optical contrast ratio with different switching times. The kinetic stability of PDT6FBT is shown in Appendix A. After switching for 2000 s, the remaining optical contrast was 10%, 10%, and 16% at 380 nm, 550 nm, and 750 nm, respectively. Electrochromic optical contrast is a significant parameter of proving cycling stability; the transmittance of the device showed a clear decrease during cycling, possibly due to the ion-trapping behavior at a high current, which led to a poor cycling stability of the electrochromic layer [43].

Response time, a critical parameter of an electrochromic material, is the time required for a 95% change in transmittance between the oxidized and reduced states, which is closely related to the following factors: the thickness and micromorphology of the polymer film, the transport ability of doping ions into and out of the polymer film during the redox process, the ionic conductivity of the electrolyte, etc. [1,2,5,6]. As shown in Table 1, the PDT6FBT film showed a fast response time (0.3–1.7 s) in both the reduction and oxidation processes at all three wavelengths. In particular, the response time during the reduction process was obviously faster than that during the oxidation process.

*CE* is an effective index for assessing the energy efficiency of electrochromic polymers. PDT6FBT exhibited comparative values at all three wavelengths with a range value of 86–97 cm^2^ C^−1^. The above electrochromic performance demonstrated that PDT6FBT was on a par with its analogues PT2BT-1F (thiophene as terminal groups), PF2BT-F (furan as terminal groups), PS2BT-F (selenophene as terminal groups), and PE2BTD-F (EDOT as terminal groups) [34,35,36].

### 2.7. Flexible Electrochromic Device

A flexible electrochromic device (ECD) was fabricated with the configuration of indium tin oxide–polyethylene terephthalate (ITO-PET)/PDT6FBT/gel electrolyte/ITO-PET. As shown in Figure 8A, the flexible ECD of PDT6FBT achieved reversible colors of navy and cyan between the neutral (0 V) and oxidized (2 V) states, consistent with the non-device phenomenon. The Δ*T* of ECD was 9% at both wavelengths, lower than that of the non-device value (Figure 8B). Electrochromic devices have a voltage drop problem due to poor packaging, uneven distribution of gel electrolytes, and uneven film thickness, resulting in different resistance in different directions. When an oxidation voltage is applied to the electrochromic device, the current is concentrated in the direction of least resistance, leading to a significant increase in the doping level and the highest transmittance. As the doping level increases, the resistance in the original direction of least resistance rises and the current is divided to other places; it is difficult to maintain the original doping level in the place with the highest transmittance at small currents; and the transmittance will decrease and show a downward trend.

### 2.8. Morphology

As shown in Figure 9, the micromorphology of PT610FBTT810, PDT6FBT, and PDT48FBT was characterized using a scanning electron microscope (SEM). At low magnification (<5000×), the three polymer films exhibited smooth surficial morphology. When the magnification reached 60,000×, more rough particles appeared on the PDT6FBT film surface than those on the PT610FBTT810 and PDT48FBT film surfaces. This may be due to PDT6FBT’s comparatively short alkyl chain, which facilitates aggregation and compact stacking.

## 3. Conclusions

In summary, three 5-fluorobenzo[c][1,2,5]-thiadiazole-based electrochromic conjugated polymers (named PT610FBTT810, PDT6FBT, and PDT48FBT) with different lengths of linear/branched alkyl chain were designed and electrosynthesized. The study demonstrated that the structure modification of alkyl chains has an obvious impact on electrochemical properties, spectroelectrochemistry, and electrochromic properties. PDT6FBT with two hexyl chain lengths showed a better electrochromic performance than PT610FBTT810 and PDT48FBT, likely due to their stronger aggregation and compact stacking. PDT6FBT displayed a distinct color change from navy to cyan upon oxidation with a 25% optical contrast ratio. Meanwhile, PDT6FBT exhibited a relatively fast response time of 0.3 s. These results indicate that 5-fluorobenzo[c][1,2,5]-thiadiazole is an appropriate unit to construct asymmetric electrochromic D-A-D-type polymers. Side-chain engineering is promising in screening high-performance electrochromic conjugated polymers, which could improve the solubility, optimize morphology, etc. The precise regulation of alkyl chains is crucial in achieving highly efficient electrochromic conducting polymers, which provides insight for designing electrochromic conducting polymers.

## Data Availability

The data are available in this publication and Appendix A.

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
