# Peer review of "Toward High-Performance Electrochromic Conjugated Polymers: Influence of Local Chemical Environment and Side-Chain Engineering"

_molecules, 2022, doi:10.3390/molecules27238424_

Round 1

Author Response

  1. The resolution of the figures (6abc and 8a) is not acceptable in the current form. Please check.

RESPONSE: We have increased the resolution of the figures.

  1. In the abstract and page 2, line 82, synthesized, not synthesised.

RESPONSE: Thank you for the reminder, we have revised it in the manuscript.

  1. Page 2, line 52, please delete end-capping to prevent misunderstanding.

RESPONSE: Thank you for your advice, we have deleted "end-capping" to avoid misunderstanding.

4.Page 2, line 59, EDOT is an electron-donating group. And the higher efficiency of which kind? Please revise to be more specific.

RESPONSE: We have revise it to be more specific, we highlighted the “coloration efficiency”.

  1. Page 2, line 72, choice for.

RESPONSE: Thank you for the reminder, we have revised it in the manuscript.

  1. Page 2 line 84, Toppare, not toppare.

RESPONSE: Thank you for the reminder, we have revised it in the manuscript.

  1. Page 3, line 119, please check the yields of each monomer; they are all lower than 80%.

RESPONSE: Thank you for your reminder, we found the previous error and have corrected it as “good yields of around 70%”.

  1. In this work, the side chain engineering is promising for screening the polymers, the authors should strengthen the importance in conclusion.

RESPONSE: Thank you for your Reminder, we have highlighted the significant of the side chain engineering for screening the polymers. “Side chain engineering is promising in screening high performance electrochromic conjugated polymers, which could improve the solubility, optimize morphology, etc.”

  1. The authors claim that PT610FBTT810 and PDT48FBT in this system, because of long alkyl chain, were soluble as soon as they were polymerized on the ITO, and it was difficult to obtain the suitable films. In this term, the measurement of polymer films should be provided with thickness measurement so the discussion between polymers could be more reasonable.

RESPONSE: The three polymers were prepared by the potentiostatic method with a charge of about 5 mC. In this electrochemical deposition condition, the thickness of polymer films was about 200 nm, which is in agreement with the reported structure[R1].

[R1] Hu, B.; Li, CY.; Liu, ZC.; Zhang, XL.; Luo, W.; Jin, L. Synthesis and multi-electrochromic properties of asymmetric structure polymers based on carbazole-EDOT and 2,5-edithienylpyrrole derivatives. Electrochim. Acta 2019, 305, 1-10.

  1. For the introduction, several works of isomeric side chain engineering, D-A-D type electrochromic polymer and polymers with asymmetric side chain should be cited.

RESPONSE: Thank you for your reminder, we have already read and cited them in the manuscript.

  1. For scheme 1, the chemical structure of 5-fluoro-2,1,3-benzothiadiazole in the monomer and polymer should be drawn as follows since there are isomer issues.

RESPONSE: We have modified the drawing of 5-fluoro-2,1,3-benzothiadiazole in Scheme 1.

Reviewer 2 Report

This manuscript reports synthesis and characterization of new electrochromic polymers and the device characteristic of one of them.  An interesting paper and should be published after minor revision about the following.

1.  Table 1 shows the response times which are different by changing the measuring wavelength.  It seems unusual and please provide explanation for this.  

2. It is difficult to find the location of the plot in Figure 6 BC. 

3. Fig. 8 B shows decreasing trend in the cycling experiment.  Please explain the mechanism.

Author Response

  1. Table 1 shows the response times which are different by changing the measuring wavelength. It seems unusual and please provide explanation for this.

RESPONSE: Response time is the time required for a 95% change in transmittance between the oxidized and reduced states at a specific wavelength. It depends on several parameters, such as the ability of the electrolyte to conduct ions as well as the ease of diffusion of these counterbalancing ionic species across the EC active layer. The transmittance is different at different wavelengths, and the degree of transmittance change is different, so it has different response times at different wavelengths. In addition, the speed of change in absorbance of the polymer at different wavelengths during doping and de-doping is different

  1. It is difficult to find the location of the plot in Figure 6 BC.

RESPONSE: We have increased the resolution of the figure.

  1. Fig.8 B shows decreasing trend in the cycling experiment. Please explain the mechanism.

RESPONSE: Electrochromic devices have a voltage drop problem due to poor packaging, uneven distribution of gel electrolyte, and uneven film thickness, resulting in different resistance in different directions. When an oxidation voltage is applied to the electrochromic device, the current will be concentrated in the direction of least resistance, leading to a significant increase in the doping level and the highest transmittance. As the doping level increases, the resistance in the original direction of least resistance rises and the current will be divided to other places, and it is difficult to maintain the original doping level in the place with the highest transmittance at small currents, and the transmittance will decrease and show a downward trend. Meanwhile, the packaging technology of the electrochromic device still needs to be improved, which will cause water and oxygen intrusion and show decreasing trend in the cycling experiment.
